# Micronutrients absorbed via the oral mucosa reduce emotion dysregulation in 5-10-year-old children: A three-phased randomized wait-list-controlled trial

**Nurina M. Katta** [1], **Neville M. Blampied**[1], **Matt Eggleston** [2], **Julia J. Rucklidge** [1] *

1 School of Psychology, Speech and Hearing, Te Whare Wānanga o Waitaha, University of Canterbury, Christchurch, New Zealand, 2 Mental Health Division, Te Whatu Ora Canterbury, Christchurch, New Zealand

* julia.rucklidge@canterbury.ac.nz

**Data Availability Statement:** Throughout the paper, we are using Figures and Tables to summarise larger amounts of data, including

## Abstract

### Objective

Previous evidence has established that micronutrient capsules can improve emotion regulation in children. This three-phased randomized open-label waitlist-controlled study investigated the safety of a micronutrient powder absorbed by the oral mucosa and its effects on emotion dysregulation in 5-to-10-year-old children. The primary outcome measures were the Revised Clinician-rated Temper and Irritability Scale (CL-ARI) and the Clinical Global Impressions-Improvement Scale (CGI-I).

### Method

Forty-eight children with moderate-to-severe symptoms of irritability were randomized to an initial treatment group (ITG) or waitlist control group (IWLG) (four-week delayed start), followed by the two groups alternating between taking the micronutrients for four weeks or having a four-week break. For the last three months of the trial, both groups took the micronutrients continuously.

### Results

Overall adherence rates were high (93%). At the end of RCT phase, there were large group differences (CL-ARI; $d = 1.25$, $p < .001$), and 67% in the ITG and 8% in the IWLG were 'much' or 'very much' improved (CGI-I). Further, the ITG displayed a clinically meaningful reduction in Attention Deficit/Hyperactivity Impulsivity Disorder (ADHD) and Oppositional Defiant Disorder (ODD) symptoms as measured with the Child Swanson, Nolan, and Pelham-IV Questionnaire 26 (SNAP-IV) compared to IWLG. The treatment effect regressed when participants stopped taking the micronutrients and was reinstated when participants were taking the micronutrients. The observed benefits were maintained over a sustained time period. The IWLG reported significantly more headaches ($p = .040$) and sweating ($p = .037$) at the end of RCT. By the end of the study, seven participants (14.5%) dropped out non-differentially by group ($p = .22$).

participant's demographic information and study results. The paper also includes a detailed methods section that outlines the study procedure, the measures used, and the way data were analysed. The Figures and Tables in the supplementary material support the data in the paper, illustrating the study procedure, additional information on side effects, and additional summarised data. The datasets generated and analysed in this study that contain the participant's raw data were conducted in form of Excel spreadsheets and spreadsheets using a statistical software (Jamovi org.). At this stage, the raw data underlying the Figures and Tables in the paper and/or supplementary material have not been provided due to ethical restrictions, so the participant's privacy is protected. The data sets contain potentially identifying and sensitive information, and since the sample recruited in the study is small, and the country the study took place in (New Zealand) has a small population, the identities of study participants could possibly be identified even if a de-identified data set was provided. During the consent procedure, participants and their caregivers were assured that confidentiality will be respected at all times and no material which could personally identify the participants would be used in any reports on this study. Further, during the consent procedure, participants were made aware that the obtained information will remain confidential and will not be disclosed or used in any way unless the researchers have concerns about the child's safety or the safety of others. Additionally, participants were told that the only people who will have access to the information are the study investigators and members of the Mental Health and Nutrition Lab Te Puna Toiora at the University of Canterbury, and approved auditors. It was also made clear that any information that was disclosed about the caregivers and the participant will be kept in a confidential file and stored in a locked filing cabinet, and that digital information will be stored on a password protected computer system, complying with Rule 5 of the Health Information Privacy Code. This ensures that the participant's information is kept safe from loss, unauthorised access, use, modification, or disclosure. Further, it was mentioned that all data that would be provided will be stored at the University of Canterbury for 10 years after the participant turns 16, and will then be destroyed, as per university and health regulations and Rule 9 of the Health Information Privacy Code. By storing the data in a different place, we would be going against this procedure as approved by the Northern Health and Disability Ethics Committee. The study and these study procedures have been approved by Northern Health and Disability Ethics

## Conclusion

The findings showed that micronutrients absorbed by the oral mucosa are a safe intervention that can effectively improve emotion dysregulation in children. Future double-blinded, randomized, placebo-controlled trials are needed to support these findings.

## Introduction

Emotion dysregulation is the maladaptive management of emotional responses that is expressed in rapid and uncontrolled shifts in emotion inappropriate to the situation, age, and developmental level [1–3]. This can interfere with fostering empathy, knowing the difference between internal emotional experiences and the external expression of emotions, and behaving socially appropriately [4, 5].

Clinically, emotion dysregulation is expressed as irritability [1], which is a response to frustration characterised by sustained grumpiness, touchiness, and resentfulness, as well as temper outbursts, intense anger, and occasional aggressive behavior [2, 6, 7]. Chronic emotion dysregulation is associated with a range of adverse psychosocial outcomes, including anxiety and depression, reduced wellbeing and self-esteem, damaged relationships, increased crime, and poor occupational and academic accomplishments [6, 8–11]. For example, the longitudinal Dunedin cohort study showed that children with undercontrolled temperament at age three were twice as likely to develop a gambling disorder at 21 and 32 relative to children classified as having a well-adjusted temperament [12]. In another study, better self-control skills in children predicted improved physical health as well as reduced substance use disorders and criminal offending in later life [13].

Typically, children make substantial progress in developing effective emotion regulation skills during the first five years of life; hence, emotion dysregulation is most predominant at preschool age and reduces over time [14, 15]. However, around 3% of youth suffer from severe irritability and clinically significant symptoms, with prevalence rates being higher in psychiatric samples [2, 6, 16].

A diagnosis where emotion dysregulation plays an important role is Attention-Deficit/Hyperactivity Disorder (ADHD), and for children with ADHD, psychiatric stimulant medication is often prescribed as the main form of treatment [1, 17, 18]. However, more than half of children with ADHD do not report improved symptoms of emotion dysregulation in response to stimulants [19, 20], and overall, the evidence for stimulants improving irritability is limited. Additionally, stimulants have been shown to sometimes increase irritability as an adverse side- or withdrawal effect [19, 21–23]. Alternatively, behavioral interventions have effectively treated emotion dysregulation [24]; however, the lack of guidelines on how to apply intervention strategies and barriers to treatment, such as transport issues, stigma, and the shortage of trained clinicians and specialist services can interfere with effective treatment delivery [25–27].

Potentially circumventing the challenges of current treatment options, recent evidence has established the role of vitamins and minerals (micronutrients) on improving emotion dysregulation. For instance, micronutrients led to a large and significant reduction in problematic social behavior and conduct problems (symptoms frequently associated with emotion dysregulation) in 14 children with ADHD [28]. Further, an RCT found that micronutrients reduced aggression, emotionally dysregulated mood and behavior, conduct problems, and hyperactivity/impulsivity and inattention to a greater degree than a placebo [29]. Another RCT found 54% of youth with both ADHD and emotion dysregulation issues who took micronutrients for

Committee (21/NTA/169). Their contact email address is hdecs@health.govt.nz. We did obtain consent from the participants that coded information may be used for future research, in which case the participants' information may be widely shared with other researchers or companies. Therefore, the minimal anonymized data set necessary to replicate the study will be made available following publication, after de-identification, on reasonable request, to researchers who provide a methodologically sound proposal and meet the criteria for access to confidential data. Proposals should be directed to the corresponding author; to gain access, data requestors will need to sign a data access agreement. An alternative point of contact is the Northern Health and Disability Ethics Committee.

**Funding:** University of Canterbury Child and Well-being Research Institute (CWRI) N.M.K https://www.canterbury.ac.nz/research/about-uc-research/research-groups-and-centres/child-wellbeing-research-institute The the sponsors or funders did not play any role in the study design, data collection and analysis, decision to publish, or preparation of the manuscript.

**Competing interests:** The authors have declared that no competing interests exist.

eight weeks improved 'much' or 'very much', while only 18% did so in the placebo group [30]. However, the product tested in these studies was micronutrient pills, and past evidence has shown that children in particular might find swallowing pills challenging [31, 32]. For example, in one study, medication was rejected in two-thirds of babies and children, or its delivery was unsuccessful [32], interfering with effective treatment.

Recently, a double-blinded RCT trial with 72 stressed university students found that micronutrients improved symptoms of irritability and anger compared to a placebo [33]. The novelty of this study was that it used a dissolvable powder absorbed in the mouth rather than capsules. The high observed adherence to treatment (95.3 to 97.3%) indicated that the powder may be a more convenient way to take micronutrients than capsules. Further, the powder is absorbed by the oral mucosa, a highly vascularized mucous membrane, which allows the micronutrients to enter the blood flow immediately. Compared to absorbing micronutrients via the gut, this mode of delivery avoids first-pass liver metabolism of the micronutrients and potentially impacts the brain more directly, potentially enhancing therapeutic effects [34, 35].

Given irritability is a prevalent issue among children that can have detrimental effects on psychosocial wellbeing, and children can struggle with swallowing capsules, further research into alternative treatment options is warranted. As emotion dysregulation typically represents normative behavior during preschool years, this study aimed at investigating the safety and effectiveness of orally-absorbed micronutrients as an intervention for 5-10-year-old children with moderate-severe symptoms of irritability.

## Methods

### Study design

This study was designed as a three-phased randomized open-label waitlist-controlled trial. All study procedures were approved by the Northern Health and Disability Ethics Committee (21/NTA/169) and prospectively registered with the Australian New Zealand Clinical Trials Registry: ACTRN12622000162718.

**Participants and entry criteria.** Participants were recruited nationally across Aotearoa New Zealand through advertisement posts on the *Te Puna Toiora* Mental Health and Nutrition Facebook page from March 2022 to June 2022. Recruitment ended when the intended sample size was reached. To be eligible, participants had to be between 5 and 10 years of age, live in New Zealand, be fluent in English, and show moderate to severe symptoms of irritability ($\geq 6$ on the Affective-Reactivity Index) [36]. Children taking any form of psychiatric medication at the time of or four weeks prior to the study were excluded from participation due to potential interactions with the micronutrients. Caregivers of participants not eligible for the trial were emailed to inform them that they were ineligible and sent information regarding local services they could approach for psychological support. See Fig 1 (CONSORT diagram) for participant flow.

**Sample size.** The number of participants was based on the What Works Clearinghouse Standards for single-case designs [37, 38], being considered a sufficient sample size to determine whether the treatment is successful in reducing emotion dysregulation with sufficient power at a 5% level of significance. The researchers anticipated a moderate effect size ($\sim d = 0.5$) based on previous studies investigating the same intervention (i.e. True Hope Ultimate Sticks), establishing moderate effect sizes on emotion dysregulation [33].

**Intervention and dosing.** The product tested is called EMPowerplus Truehope Ultimate Sticks (Sticks), which are pouches containing 36 vitamins, minerals, and amino acids in form of a powder and available in banana, sour berry, and tropical punch flavor. Participants were asked to take one Stick per day, putting the micronutrient powder in their mouth and waiting

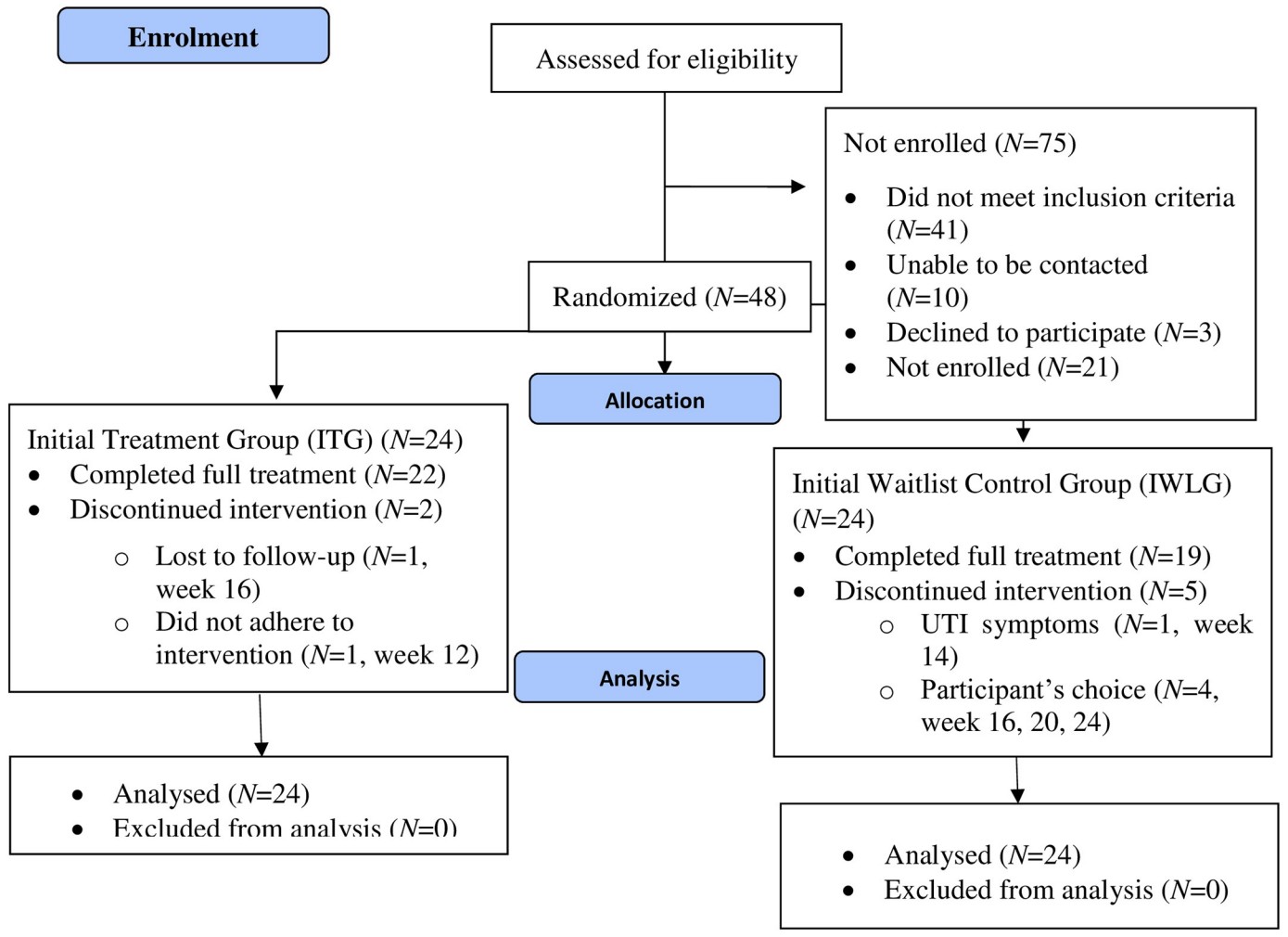

**Fig 1. CONSORT flow diagram.**

until it was dissolved. It was suggested to not eat or drink anything 10 minutes before and after taking the intervention. The ingredients of the Sticks are shown in S1 Table in the supplemental material.

**Study phases.** *Screening and consent procedure.* Potential participants were screened for eligibility by obtaining information regarding their demographics as well as completing measures assessing their children's emotion dysregulation via an online questionnaire. If children were eligible, participants and their caregivers were met face-to-face or via Zoom to explain the study procedure and, if participants wanted to take part, obtain written informed consent from the caregivers and assent from the children.

*Baseline phase.* Caregivers completed a baseline assessment online and met with the study coordinator in person at the University of Canterbury or online via Zoom together with their child to complete a variety of clinician-administered measures (see measures below).

*RCT phase.* Once the baseline assessment had been completed and the participants had been randomized and allocated to either the initial treatment group (ITG) or initial waitlist group (IWLG), they were asked to take the intervention for four weeks or told that they had a four-week delayed start to taking the intervention, respectively.

*Immediate Replication Phase (IRP)*. The IRP used a parallel groups design to detect if potential treatment effects observed in the RCT phase could be replicated and whether accumulation of therapy effects could be detected. During the IRP phases (week 8 and 12), the groups took turns at alternately taking and then stopping the micronutrients every four weeks.

*Treatment maintenance phase*. The treatment maintenance phase examined whether the potential treatment effect was maintained throughout a further two-month period. At the end of the maintenance phase, participants in both groups had taken the micronutrients for three months continuously. Study phases are shown in S1 Fig in the supplemental material.

## Outcome measures

Clinician-rated measures were completed by the study coordinator who met with the participants and their caregivers face-to-face or by Zoom. All other questionnaires were completed by the caregiver online.

**Primary outcome measures.** *Revised Clinician-Rated Temper and Irritability Scale (CL-ARI)*. The CL-ARI is a clinician-administered measure that assesses irritability in children over the past week, considering temper outbursts and irritable mood, their frequency, severity, and duration, as well as potential impairment in the family, in school, or with peers over three subscales: Temper Outbursts (e.g. "Did your child show any of these moderate verbal or physical behaviors over the past week?"), Irritable Mood (e.g. "Thinking now about irritable mood in particular, over the past week how many days was your child's mood negative, cranky, grumpy, or crabby?"), and Impairment ("To what extent did your child's temper and/or irritable mood cause problems for the family?"). The items are scored on a 5-point Likert scale (e.g. 0 = none, 4 = more than one outburst every day). Total CL-ARI scores range from 0–100, with higher scores indicating more severe emotion dysregulation. The scale has good internal consistency ($\alpha$ = .89) and adequate test-retest reliability ($ICC$ = .67) [39]. Caregivers were interviewed to collect data every four weeks on the Total CL-ARI and all subscales. The CL-ARI was administered by the study coordinator, a postgraduate PhD student in Psychology and supervised by a clinical psychologist. As part of the supervision, the supervisor conducted reliability checks and assessed whether the CL-ARI procedure was followed correctly.

*Clinical Global Impressions—Improvement Scale (CGI-I) [40]*. The CGI-I is a clinician-administered scale that rates potential changes in the participants' functioning compared to baseline based on all information obtained from the caregivers. Each item has seven responses, ranging from 'very much improved' (1) to 'very much worse' (7) and helps to provide a brief, overall assessment of the participant. The CGI-I has demonstrated high external validity ($r$ = 0.74) [41]. Aside from baseline, the CGI-I was completed every four weeks. At the end of each treatment phase, participants were classified as 'Responders' if they were identified as having 'much' or 'very much' improved, based on all information that was available to the study coordinator, including all parent-report and clinician-administered measures. The CGI-I was administered every four weeks by the study coordinator and supervised by a clinical psychologist.

**Secondary outcome measures.** *Parent-Target-Problem Rating Scale (PTP)*. The PTP is a clinician-administered measure that asks the caregiver to identify two main issues the child struggles with. Once identified, the clinician asked follow-up questions regarding some examples of the problem, as well as its frequency, duration, and intensity. Every four weeks, the study coordinator assessed changes in the identified problem behaviors, rating each on a 9-point Likert scale (1 = problem resolved or extremely improved, 9 = extremely worse).

*Clinical Global Impressions—Severity Scale (CGI-S) [40]*. The CGI-S is a clinician-administered measure that rates the participant's severity of symptoms, functioning, and behavior over

the past seven days. The scale asks "Considering your total clinical experience with this particular population, how impaired is the patient at this time?" The item is scored on a 7-point Likert scale from 1 (normal, not impaired) to 7 (markedly impaired) [40]. The CGI-S was administered by the study coordinator and supervised by a clinical psychologist.

*Children's Global Assessment Scale (CGAS) [42].* The CGAS is a clinician-reported measure that assesses children's overall functioning based on the observed and reported information at each visit (every four weeks). The scale ranges from 0 to 100, with higher scores representing better functioning, for example: "60–51: Variable functioning with sporadic difficulties or symptoms in several but not all social areas; disturbance would be apparent to those who encounter the child in a dysfunctional setting or time but not to those who see the child in other settings."

*Affective Reactivity Index (ARI) [36].* The ARI is a concise, caregiver-rated scale assessing the child's irritability, incorporating seven items asking about feelings of irritability during the past seven days (e.g., "My child is easily annoyed by others", "My child gets angry frequently", "Overall, irritability causes my child problems"). Statements can be scored from 0 (not true) to 2 (certainly true). The ARI has been shown to have good internal consistency ($\alpha$ = .84 -.92) [43] and is considered an appropriate measure to assess irritability [36]. The total score is the sum of the first six items and ranges from 0–12. The ARI was completed online every two weeks by the caregiver.

*Emotional Outburst Inventory (EMO-1) [44].* The EMO-I is a caregiver-rated irritability measure that assesses phasic irritability in youth in clinical settings, assessing outburst severity, frequency, and duration. It has adequate internal consistency ($\alpha$ = 0.82) and validity. It has nine questions including multiple-choice items, Likert-scale items, and one open question ("What helps your child to calm down?"). The EMO-1 was used to describe the sample at baseline and at the end of the trial in terms of outburst behaviors, frequency, and duration.

*Strengths and Difficulties Questionnaire (SDQ) [45].* The SDQ has 25 items, each scored on a 3-point Likert scale ('not true', 'somewhat true', and 'certainly true') across five subscales (emotional symptoms, conduct problems, hyperactivity/inattention, peer relationship problems, and prosocial behavior) in 4–17 year old children and adolescents [46]. Scores in the normal range lie between 0–13 whereas scores between 17–40 represent severe psychological symptoms. The SDQ also assesses how severely the observed symptoms interfere with family relationships, friendships, learning in the classroom, and leisure activities. This 'impact score' is assessed by means of a 0 to 4 Likert scale ('not at all' to 'a great deal'), with a score of 2 or greater representing problems in such areas [28]. The SDQ was completed by the caregiver every four weeks.

*The Child Swanson, Nolan, and Pelham-IV Questionnaire 26 (SNAP-IV 26).* The SNAP-IV 26 is a shorter version of the long SNAP-IV [47] and measures symptoms of Attention-Deficit/ Hyperactivity Disorder (ADHD) and Oppositional Defiant Disorder (ODD). It is a caregiver-rated scale with three subscales: inattention (items 1–9), hyperactivity-impulsivity (items 10–18), and ODD symptoms (items 19–26), each scored on a 4-point Likert scale (0 = not at all, 1 = just a little, 2 = quite a bit, and 3 = very much), with higher scores representing more symptom severity. For the inattention, hyperactivity/impulsivity and ODD subscale, scores of less than 13, 13, and 8 represent normal symptoms, while scores from 23–27, 23–27, and 19–24 represent severe symptoms, respectively. The SNAP-IV displays high internal consistency ($\alpha$ = .94) and sufficient interrater reliability between teachers and caregivers ($r$ = .43-.49) [48].

*Eating behavior questionnaire [49].* Caregivers were asked to report on their child's diet and eating pattern every four weeks using a 5-point Likert scale, rating servings of fruit and vegetables eaten/day (1 = less than one serving a day, 5 = 4 or more servings per day), how much food is eaten/meal, regularity of eating breakfast, and consumption frequently of processed

and fast-food (5 = every day, 1 = less than once a week). The child's diet overall is rated from 1 (not very healthy) to 7 (very healthy). The total score ranges from 9 to 47, with a higher score representing a healthier diet.

*The Depression Anxiety and Stress Scale 21 (DASS-21)*. The DASS-21 [50] is a publicly available questionnaire that has three subscales (depression, anxiety, and stress), using a 3-point Likert scale. Cut-offs have been provided to indicate normal, mild, moderate, severe, or extremely severe problems; anything below 10 (for depression), 8 (for anxiety) and 15 (for stress) is considered within the normal to mild range. The scale has excellent reliability (*ICC* = .93) as well as a sufficient construct validity [51]. The DASS-21 was completed by the caregivers about their own mental health to capture their symptoms every four weeks.

*The revised Side-Effect Checklist (SEC)*. Safety, or side effects were assessed using the SEC, a questionnaire based on the Antidepressant Side-Effect Checklist (ASEC), which is normally a self-report measure looking at adverse events associated with antidepressants [52]. The SEC was modified to assess common side effects associated with taking nutritional supplement capsules, such as nausea, headaches, or insomnia. The SEC includes 21 items that are scored on a 0- to 3-point Likert scale, investigating the type, presence, and severity of multiple side effects that may be related to the intervention. However, in this study, the SEC was completed by the participant's caregiver. They were also asked to indicate whether they thought that the experienced side effect was related to the intervention or was due to other reasons.

*Adherence and product likability*. Every two weeks, caregivers reported how many doses the participants missed ("How many doses did your child miss over the past two weeks? One dose is equal to one Stick") and rated on a Likert scale how easy it was to take the product (1 = extremely difficult, 7 = extremely easy) and how much the Sticks were liked by the children (1 = dislike a great deal, 7 = like a great deal) on behalf of their child.

## Randomization (random number generation, allocation concealment, implementation) and blinding

Participants were randomized in blocks of four to the initial treatment and initial waitlist control group (ITG and IWLG respectively) using a computer-generated randomization sequence (from www.randomization.com) by an independent person not involved in the recruitment of participants. The website asked about the set of numbers that needed to be generated, the numbers per set, and the number range. Once the randomization list had been generated, participants were assigned to the next sequential number that had been written on a paper and concealed in envelopes by an independent person not involved in the recruitment process. During the baseline meeting, the envelopes were opened by the study coordinator in front of the participants, who had been blinded to participant randomization and assignment up to this point.

## Data analysis

Data were analysed using Jamovi 2.2.5 (jamovi.org), software by Lakens [53] and Cumming, Fidler [54] for effect sizes (ES) and 95% Confidence Intervals (95%CI) on ES, and Sigma Plot (systatsoftware.com). Demographic data were tabulated descriptively. Standardized mean difference effect sizes were reported as Cohen's $d_s$ for between-groups and as $d_{av}$ for within-group comparisons (with 95%CI on *d*) (Lakens, 2103). The Common Language ES (CLES) [53] was additionally reported for the CL-ARI at the end of RCT, giving the probability that a randomly selected individual in the ITG improved relative to a randomly selected IWLG participant. At baseline, the variables were tested for normality using the Shapiro-Wilk normality test, and all variables that failed the normality test were analysed using the Welch's test, a

parametric test that controls for homogeneity of variance. Remaining mean group differences in the RCT and IRP were analysed by two-sided *t*-tests at each time point while within-group mean changes were compared with *t*-tests comparing baseline with later time points. During treatment maintenance phases, change over time and between-group differences were examined by a Groups (2) by Time (2) ANOVA. For all statistical tests α was set to.05. Correlations were conducted using Pearson's *r*.

Modified Brinley Plots [55] examined the patterns of response to treatment over study phases, displaying individual change across time in the context of all other participants' scores at that time, along with means (*M*), standard deviations (*SD*), and relevant effect sizes.

# Results

## Demographic information

All participants completed the RCT phase, 22 (91.7%) the full trial in the ITG, and 19 (79.2%) in the IWLG. Reasons for drop-out are shown in Fig 1. Both groups were predominantly of NZ European ethnicity and closely matched on age, but the ITG was more male dominated than the IWLG group (Table 1). In both groups, participants had a moderate-to-high socioeconomic status (SES) as determined by the New Zealand Socio-economic Index 2018 [56]. Primary ethnicity was categorized using the New Zealand Ministry of Health prioritization tables based on the Ethnicity New Zealand Standard Classification 2005 V2.0 [57].

**Table 1. Sociodemographic characteristics of participants at baseline.**

| Baseline characteristic | ITG (*N* = 24) | | IWLG (*N* = 24) | |
|---|---|---|---|---|
| | *N* | % | *N* | % |
| Age | 7.9 (*M*) | 1.76 (*SD*) | 7.7 (*M*) | 1.5 (*SD*) |
| Gender | | | | |
| Female | 7 | 29.2 | 10 | 41.2 |
| Male | 17 | 70.8 | 13 | 54.2 |
| Non-binary | 0 | 0 | 1 | 4.2 |
| Ethnicity | | | | |
| NZ Māori | 4 | 16.7 | 4 | 16.7 |
| Pacific Peoples | 0 | 0 | 0 | 0 |
| Asian | 2 | 8.3 | 0 | 0 |
| MELAA | 0 | 0 | 1 | 4.2 |
| Other | 2 | 8.3 | 2 | 8.3 |
| NZ European | 16 | 66.7 | 17 | 70.8 |
| SES | 66.1 (*M*) | 11.1 (*SD*) | 64.8 (*M*) | 15.3 (*SD*) |
| Diagnosis [a] | | | | |
| Anxiety Disorders | 3 | 6.3 | 5 | 10.4 |
| Behavioral Problems | 5 | 10.4 | 7 | 14.6 |
| Learning Difficulties | 3 | 6.3 | 4 | 8.3 |
| Previous psychological treatment [a] | 6 | 25.0 | 4 | 16.7 |
| Previous psychotropic medication [a] | 1 | 4.2 | 1 | 4.2 |
| Current psychological treatment [a] | 1 | 4.2 | 4 | 16.7 |

*Note*. MELAA = Middle Eastern, Latin American, and African; SES = Socio-economic status

[a] based on parent-report

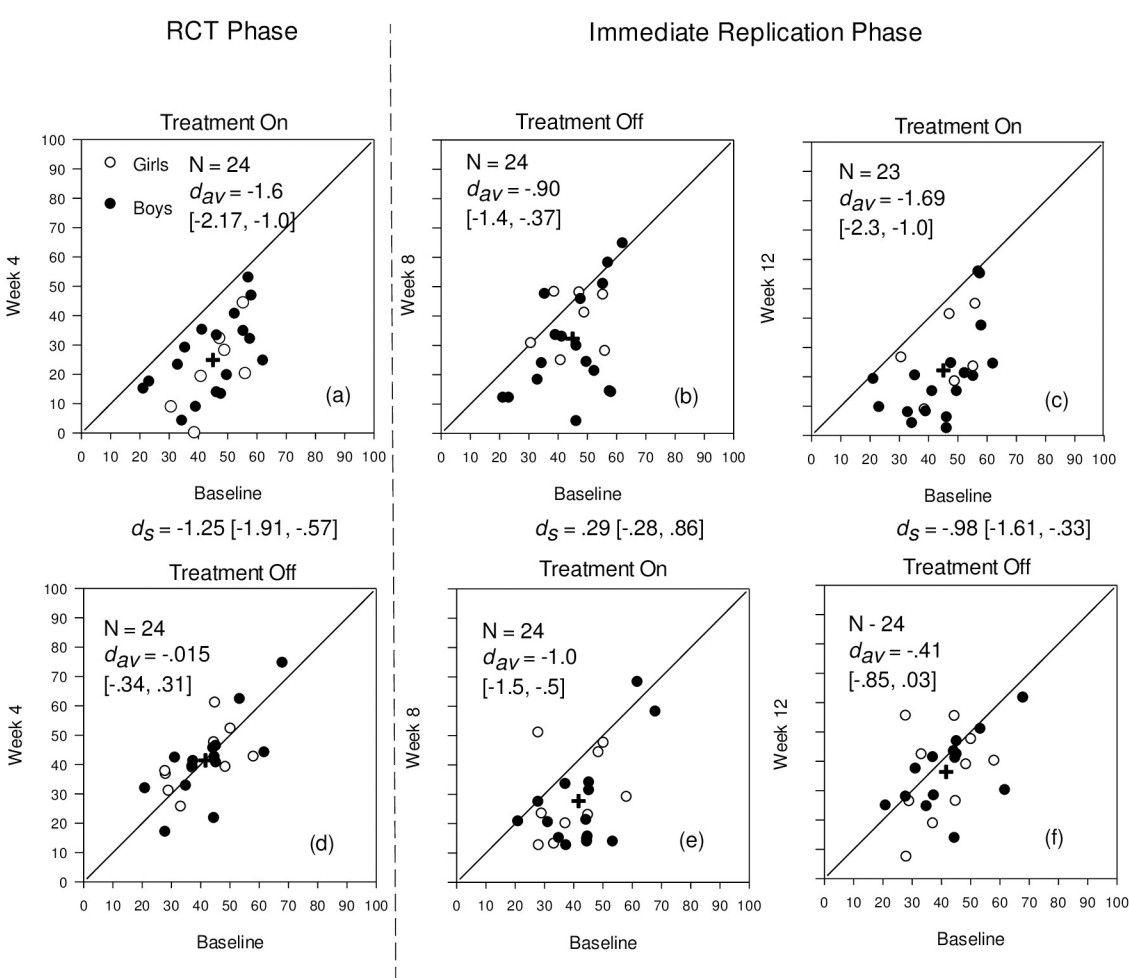

**Fig 2. Changes for the ITG and IWLG on the CL-ARI from baseline to week 4, 8, and 12.** Note. Modified Brinley Plots showing individual scores and average change (as shown with the cross) at baseline and at week 4, 8, and 12 for the Revised Clinician-Rated Temper and Irritability Scale (CL-ARI). The solid diagonal line indicates no change. The between- ($d_s$) and within-group ($d_{av}$) effect sizes are shown at each time point. The 95% Confidence Interval is shown in [] .

## Randomized Controlled Phase (RCT) and Immediate Replication Phases (IRP)

Results for the CL-ARI between groups and across time are shown in Fig 2 and group data for the primary and secondary measures are displayed in Table 2.

**Primary outcomes.** *CL-ARI.* As shown in Figs 2 and 3, both the ITG and IWLG displayed similar scores and variance baseline. At the end of RCT phase, the ITG largely improved their total score ($t(23)$ = -8.54, $p < .001$, $d_{av}$ = 1.6) while the IWLG showed negligible change ($t(23)$ = -0.09, $p = .0 = 926$, $d_{av}$ = -0.015). At the end of RCT phase, the ITG improved largely and statistically significantly more than the IWLG ($t(46)$ = -4.33, $p < .001$, $d_s$ = 1.25). Further, at the end of the RCT phase, the CLES was large (81%), indicating a high probability of severe symptom reduction in the treatment group. This pattern of improvement is also evident for the CL-ARI subscales.

As displayed in Fig 2, for the ITG, going off the micronutrients (week 8) did not lead to a systematic return to baseline levels on the CL-ARI, although there was a reduction in $d_{av}$ ($t(23)$

**Table 2. Between group baseline and week 4, 6, 8, and 12 data on primary and secondary outcome measures.**

| Variable | Baseline (N = 48) | | Week 4 (RCT) (N = 48) | | | | Week 8 (IRP) (N = 48) | | | | Week 12 (IRP) (N = 47) | | | |
|---|---|---|---|---|---|---|---|---|---|---|---|---|---|---|
| | ITG | IWLG | ITG | IWLG | | | ITG | IWLG | | | ITG (N = 23) | IWLG (N = 24) | | |
| | M (SD) | M (SD) | M (SD) | M (SD) | $d_s$ | p | M (SD) | M (SD) | $d_s$ | p | M (SD) | M (SD) | $d_s$ | p |
| CL-ARI[+] | | | | | | | | | | | | | | |
| Temper Outbursts | 0.44 (0.12) | 0.46 (0.10) | 0.30 (0.17) | 0.42 (0.12) | -0.82 | .006* | 0.35 (0.13) | 0.33 (0.15) | 0.12 | .673 | 0.29 (0.18) | 0.40 (0.12) | -0.71 | .019* |
| Irritable Mood | 0.56 (0.22) | 0.48 (0.22) | 0.30 (0.20) | 0.53 (0.20) | -1.19 | <.001* | 0.42 (0.28) | 0.33 (0.24) | -0.36 | .224 | 0.26 (0.22) | 0.45 (0.23) | -0.83 | .007* |
| Impairment | 0.365 (0.15) | 0.31 (0.17) | 0.15 (0.11) | 0.29 (0.16) | -1.01 | .001* | 0.20 (0.16) | 0.18 (0.13) | 0.18 | .532 | 0.12 (0.12) | 0.25 (0.17) | -0.88 | .004* |
| Total | 44.9 (11.3) | 41.6 (11.7) | 24.9 (13.8) | 41.4 (12.7) | -1.25 | <.001* | 32.2 (16.3) | 27.7 (15.7) | 0.29 | .316 | 22.2 (15.4) | 36.4 (13.7) | -0.98 | .002* |
| CGI-I[+] | | | 2.33 (0.76) | 3.88 (0.99) | 1.74 | <.001* | 3.17 (1.17) | 2.79 (1.10) | 0.33 | .258 | 2.39 (0.78) | 3.64 (1.17) | -1.23 | <.001* |
| CGI-S | 3.96 (0.55) | 4.00 (0.65) | 2.92 (1.02) | 3.92 (0.66) | -1.17 | <.001* | 3.42 (0.88) | 3.38 (0.77) | 0.05 | .862 | 2.88 (1.08) | 3.83 (0.76) | -1.03 | <.001* |
| CGAS | 55.1 (6.33) | 55.3 (6.55) | 64.2 (9.50) | 55.8 (7.09) | 1.00 | .001* | 60.2 (8.92) | 61.0 (8.09) | -0.10 | .736 | 65.1 (9.11) | 57.4 (7.62) | 0.92 | .003* |
| PTP | | | 3.21 (1.32) | 4.71 (0.91) | -1.33 | <.001* | 4.08 (1.41) | 3.63 (1.28) | 0.34 | .245 | 3.13 (1.25) | 4.38 (1.17) | -1.03 | .001* |
| ARI | 8.38 (1.84) | 8.38 (2.02) | 3.67 (2.32) | 7.08 (2.0) | -1.58 | <.001* | 5.75 (2.98) | 4.83 (3.02) | 0.31 | .295 | 3.30 (2.34) | 5.79 (3.11) | -0.90 | .003* |
| SDQ | | | | | | | | | | | | | | |
| Externalising Score | 11.0 (2.77) | 11.7 (4.33) | 8.85 (2.95) | 11.6 (4.07) | -0.84 | .006* | 9.29 (3.32) | 9.79 (4.80) | -0.121 | .677 | 7.96 (3.46) | 10.33 (4.34) | -0.60 | .044* |
| Internalising Score | 5.88 (3.13) | 6.96 (4.14) | 3.58 (2.55) | 7.08 (3.46) | -1.15 | <.001* | 4.42 (3.73) | 5.42 (3.17) | -0.29 | .323 | 2.87 (2.75) | 6.04 (3.78) | -0.10 | .002* |
| Total Difficulties Score | 16.6 (3.93) | 18.6 (7.13) | 12.2 (4.1) | 18.7 (5.89) | -1.28 | <.001* | 13.71 (5.19) | 15.21 (6.55) | -0.254 | .384 | 10.83 (4.67) | 16.38 (6.76) | -0.95 | .002* |
| Impact Score | 3.63 (1.66) | 3.92 (1.91) | 2.21 (1.59) | 3.25 (2.21) | -0.54 | .068 | 2.42 (1.38) | 2.83 (2.41) | -0.212 | .467 | 1.26 (1.52) | 3.38 (2.83) | -0.93 | .003* |
| SNAP-IV | | | | | | | | | | | | | | |
| ADHD | 29.6 (10.2) | 33.0 (12.5) | 22.6 (11.1) | 29.3 (13.1) | -0.55 | .062 | 24.2 (11.7) | 26.9 (13.3) | -0.21 | .465 | 22.1 (10.5) | 29.8 (14.8) | -0.60 | .046* |
| ODD | 15.5 (4.88) | 15.1 (4.74) | 10.0 (4.74) | 13.6 (4.11) | -0.82 | .007* | 11.0 (5.20) | 11.50 (5.73) | -0.08 | .773 | 9.04 (5.60) | 12.7 (5.30) | -0.66 | .027* |
| EBQ | 34.8 (5.47) | 34.4 (5.95) | 35.7 (4.85) | 34.3 (5.70) | 0.27 | .059 | 34.9 (5.73) | 34.1 (5.33) | 0.14 | .641 | 35.0 (5.70) | 34.4 (5.72) | 0.10 | .728 |
| DASS-21 | | | | | | | | | | | | | | |
| Depression | 4.67 (5.49) | 7.08 (6.49) | 3.83 (3.82) | 4.17 (5.17) | -0.07 | .800 | 2.92 (2.83) | 4.75 (7.39) | -0.33 | .265 | 2.26 (2.12) | 5.67 (6.15) | -0.74 | .016* |
| Anxiety | 2.58 (2.73) | 4.42 (5.66) | 1.83 (2.76) | 3.00 (4.04) | -0.34 | .250 | 2.00 (2.57) | 2.75 (3.58) | -0.24 | .409 | 1.83 (2.82) | 3.17 (3.77) | -0.40 | .174 |
| Stress | 13.50 (7.42) | 14.50 (6.76) | 9.25 (5.59) | 11.25 (6.97) | -0.32 | .279 | 10.83 (6.40) | 10.17 (6.27) | 0.11 | .717 | 10.00 (6.06) | 13.42 (9.06) | -0.44 | .138 |

Note. M = mean; SD = standard deviation, d = Cohen's d effect size; RCT = randomized control trial phase; IRP = immediate replication phase; CL-ARI = Revised Clinician-rated Affective Reactivity Scale; CGI-I = Clinical Global Impressions Improvement Scale; PTP = Parent Target Problem Rating Scale; ARI = Affective Reactivity Index; SDQ = Strengths and Difficulties Questionnaire; SNAP-IV = The Child Swanson, Nolan, and Pelham-IV Questionnaire 26; EBQ = Eating Behavior Questionnaire.

[+] = primary measure.

* p < .05.

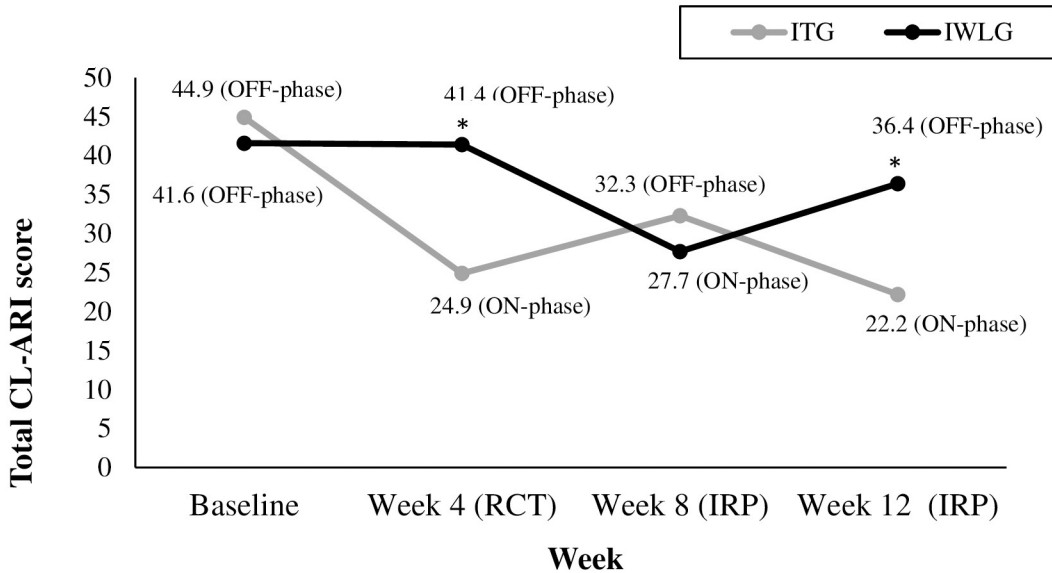

**Fig 3. CL-ARI total mean scores for the ITG and IWLG across baseline and week 4, 8, and 12.** Note. This Fig presents the Total Revised Clinician-Rated Temper and Irritability Scale (CL-ARI) scores for the ITG and IWLG across baseline until the end of the immediate replication phase (IRP)) of the trial. * $p < .05$ (between-group comparison).

= -3.91, $p < .001$, $d_{av}$ = .90). Receiving treatment for the first time in the IWLG (week 8) led to a similar but attenuated effect (Fig 2a and 2e), ($t(23)$ = -4.84, $p < .001$, $d_{av}$ =.-1.0). The between-group $d_s$ at week 8 was small and statistically not significant ($t(46)$ = 1.01, $p$ = .32, $d_s$ = .29). Return to micronutrients for the ITG (week 12) replicated the initial positive effect with a large statistically significant effect ($t(22)$ = -8.28, $p < .001$, $d_{av}$ = 1.69) (Fig 2a and 2c), while withdrawal of treatment for the IWLG led to a trend towards the retention of a treatment effect into the OFF-phase, with a small-to-moderate ES ($t(23)$ = -1.92, $p$ = .067, $d_{av}$ = -.41).

*CGI-I.* The ITG was rated as significantly and largely more improved than the IWLG at the end of RCT ($t(46)$ = -6.04, $p < .001$, $d_s$ = 1.74). In addition, 66% ($N$ = 16) of the ITG were identified as *responders* to the treatment ('much-' or 'very much improved'), while only 8% ($N$ = 2) 'much' or 'very much' improved in the IWLG group. Going off treatment for the ITG lead to 33% ($N$ = 8) responders relative to baseline, while starting treatment for the IWLG resulted in 54% responders ($N$ = 13) (week 8). Upon continuing the micronutrients, 60% ($N$ = 14) of the ITG responded to treatment, while going off treatment led to 21% ($N$ = 5) of responders in the IWLG compared to baseline.

**Secondary outcomes.** Data for the secondary outcome variables are shown in Table 2.

*CGI-S.* At the end of RCT, the ITG was functioning significantly better than the IWLG at the end of RCT (*Welch's* $t(39.2)$ = -4.05, $p < .001$, $d_s$ = 1.17). The ITG mean fell to within the 'mildly ill' range, while the IWLG mean stayed within the 'moderately ill' range. Further, nine (37.5%) participants were identified as 'not at all ill' or 'borderline ill' in the ITG versus none in the IWLG.

*CGAS.* At the end of RCT, the ITG had large and significantly better overall functioning than the IWCG ($t(46)$ = 3.46, $p < .001$, $d_s$ = 1.00). Specifically, at the end of RCT, the CGAS scores in the ITG represented "some difficulty in a single area but generally functioning well" ($M$ = 64.2, $SD$ = 9.50) compared to the IWLG, which indicated "variable functioning with sporadic difficulties in several but not all social areas" ($M$ = 55.8, $SD$ = 7.09).

**Table 3. Results for treatment maintenance phases on primary and secondary measures.**

| Variable | Baseline | | Month 1 | | Month 2 | | $\eta_p^2$ (between) | $p$ (between) | $\eta_p^2$ (within) Month 1–2 | $p$ (within) Month 1–2 |
|---|---|---|---|---|---|---|---|---|---|---|
| | ITG (N = 24) M (SD) | IWLG (N = 24) M (SD) | ITG (N = 22) M (SD) | IWLG (N = 20) M (SD) | ITG (N = 22) M (SD) | IWLG (N = 19) M (SD) | | | | |
| CL-ARI[+] | | | | | | | | | | |
| Temper Outbursts | 0.44 (0.12) | 0.46 (0.10) | 0.25 (0.19) | 0.33 (0.14) | 0.23 (0.18) | 0.30 (0.18) | 0.07 | .103 | .04 | .233 |
| Irritable Mood | 0.56 (0.22) | 0.48 (0.22) | 0.25 (0.26) | 0.37 (0.24) | 0.28 (0.25) | 0.37 (0.27) | 0.06 | .123 | .00 | .911 |
| Impairment | 0.365 (0.15) | 0.31 (0.17) | 0.12 (0.11) | 0.20 (0.21) | 0.14 (0.16) | 0.21 (0.19) | 0.07 | .102 | .014 | .460 |
| Total | 44.9 (11.3) | 41.6 (11.7) | 20.2 (16.6) | 29.7 (17.0) | 21.3 (18.3) | 29.5 (18.9) | 0.08 | .080 | 0.00 | .981 |
| CGI-I[+] | | | 2.27 (0.88) | 3.05 (1.32) | 2.27 (0.94) | 3.16 (1.34) | 0.15 | .011* | 0.00 | 1.00 |
| CGI-S | 3.96 (0.55) | 4.00 (0.65) | 2.77 (1.15) | 3.45 (1.10) | 2.64 (1.26) | 3.16 (0.96) | 0.10 | .049* | 0.14 | .015* |
| CGAS | 55.1 (6.33) | 55.3 (6.55) | 66.2 (10.5) | 61.3 (9.58) | 68.2 (13.4) | 62.8 (8.88) | 0.07 | .088 | 0.19 | .005* |
| PTP | | | 2.86 (1.17) | 3.70 (1.69) | 2.82 (1.33) | 3.37 (1.42) | 0.08 | .080 | 0.05 | .147 |
| ARI | 8.38 (1.84) | 8.38 (2.02) | 3.25 (2.61) | 5.00 (3.16) | 2.82 (3.00) | 4.37 (3.08) | 0.09 | .065 | 0.20 | .004* |
| SDQ | | | | | | | | | | |
| Externalising Score | 11.0 (2.77) | 11.7 (4.33) | 7.52 (3.67) | 8.80 (3.79) | 6.86 (3.58) | 8.53 (3.88) | 0.04 | .206 | 0.09 | .062 |
| Internalising Score | 5.88 (3.13) | 6.96 (4.14) | 3.13 (2.83) | 5.00 (3.43) | 2.32 (2.08) | 4.63 (3.37) | 0.13 | .022* | 0.11 | .036* |
| Total Difficulties Score | 16.6 (3.93) | 18.6 (7.13) | 10.7 (4.82) | 13.8 (6.27) | 9.18 (4.98) | 13.2 (5.93) | 0.10 | .040* | 0.20 | .003* |
| Impact Score | 3.63 (1.66) | 3.92 (1.91) | 1.39 (1.59) | 2.15 (2.41) | 1.00 (1.48) | 2.00 (2.47) | 0.06 | .140 | 0.07 | .086 |
| SNAP-IV | | | | | | | | | | |
| ADHD | 29.6 (10.2) | 33.0 (12.5) | 22.0 (11.3) | 23.1 (11.2) | 21.0 (11.1) | 25.4 (12.4) | 0.02 | .409 | 0.02 | .421 |
| ODD | 15.5 (4.88) | 15.1 (4.74) | 8.17 (5.37) | 9.95 (5.24) | 8.86 (6.88) | 9.95 (5.19) | 0.02 | .382 | 0.00 | .776 |
| EBQ | 34.8 (5.47) | 34.4 (5.95) | 35.3 (5.64) | 34.8 (5.20) | 36.3 (5.61) | 35.1 (5.01) | 0.01 | .576 | .030 | .276 |
| DASS-21 | | | | | | | | | | |
| Depression | 4.67 (5.49) | 7.08 (6.49) | 2.70 (2.93) | 4.70 (7.00) | 2.27 (3.40) | 4.32 (5.34) | 0.06 | .128 | 0.02 | .388 |
| Anxiety | 2.58 (2.73) | 4.42 (5.66) | 1.65 (3.60) | 3.60 (6.34) | 2.18 (3.75) | 2.74 (5.26) | 0.02 | .375 | 0.01 | .469 |
| Stress | 13.50 (7.42) | 14.50 (6.76) | 9.30 (6.54) | 10.7 (8.24) | 8.91 (8.72) | 10.1 (5.55) | 0.01 | .548 | 0.003 | .732 |

Note. M = mean; SD = standard deviation, d = Cohen's d effect size measure; RCT = randomized control trial phase; IRP = immediate replication phase;

CL-ARI = Revised Clinician-rated Affective Reactivity Scale; CGI-I = Clinical Global Impressions Improvement Scale; PTP = Parent Target Problem Rating Scale;

ARI = Affective Reactivity Index; SDQ = Strengths and Difficulties Questionnaire; SNAP-IV = The Child Swanson, Nolan, and Pelham-IV Questionnaire 26;

EBQ = Eating Behavior Questionnaire.

[+] = primary measure

* $p < .05$

*PTP*. At the end of RCT, 67% (N = 16) of participants in the ITG had 'definitely', 'much', or 'extremely improved' in terms of emotion dysregulation symptoms, including anger, temper, frustration, or emotional outbursts. In contrast, only 8% (N = 2) in the IWLG were rated as such. Overall, at the end of RCT, the treatment group average fell in the 'definitely improved' range, while the IWLG mean fell in the 'no change from baseline'.

*DASS-21*. At the end of RCT, a positive correlation between scores on the Total CL-ARI and Total DASS-21 was found (r = 0.38, p = .007), indicating that the lower the participants' symptoms of irritability, the lower the caregivers' symptoms of depression, anxiety, and stress, and vice versa.

## Treatment maintenance phase

See Table 3 for all maintenance phase results on the primary and secondary measures.

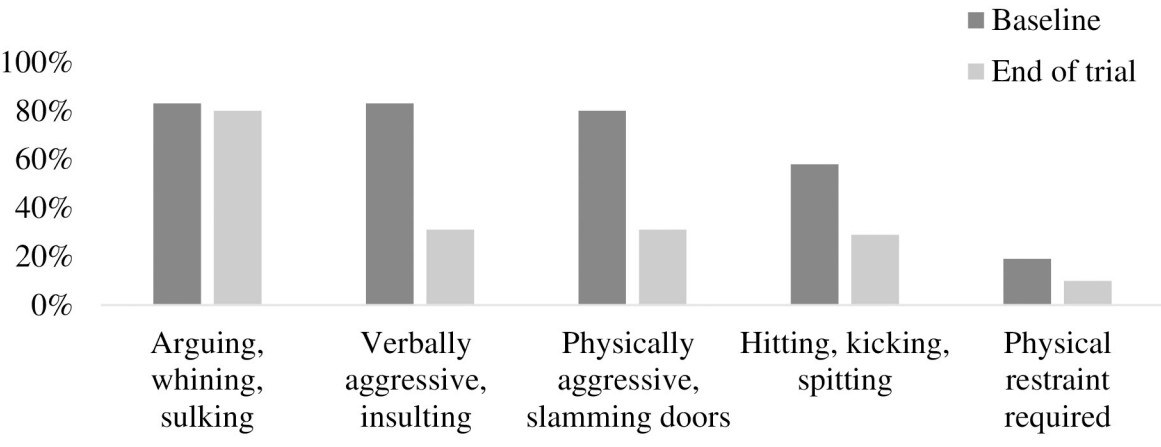

**Fig 4. Descriptive information (%) for the EMO-I for participants at baseline and end of trial.**

**Primary outcomes.** *CL-ARI.* No significant group-, time-, or interaction effects were found on the total CL-ARI or any of the subscales in the maintenance phase.

*CGI-I.* Post-hoc analysis showed that across the treatment maintenance phase, the ITG showed a significantly greater improvement over both month 1 and 2 relative to the IWLG, although the difference was small ($t(39) = -2.66$, $p = .011$, $d = 0.15$). While in month 1, 65% ($N = 15$) responded to treatment in the ITG, 45% ($N = 9$) did so in the IWLG. Similarly, at month 2, 68% ($N = 15$) in the ITG and 47% ($N = 9$) in the IWLG responded to treatment.

**Secondary outcomes.** *CGI-S.* At the end of the trial, participants in the ITG functioned slightly but significantly better than the IWLG ($F(1,39) = 6.43$, $p = .015$, $\eta_p^2 = 0.10$). Further, 55% ($N = 12$) of the ITG were rated 'not at all ill' or 'borderline ill', while 32% ($N = 6$) were rated as such in the IWLG.

*PTP.* In month 1, 72.7% ($N = 16$) definitely- much-, or extremely improved in the ITG regarding parent-identified emotion dysregulation-related issues, while 50% did so in the IWLG. In month 2, 68% ($N = 15$) were classified as such in the ITG, and 47% in the IWLG.

*EMO-I.* As shown in Fig 4, at baseline, 83% of children argued, whined or sulked, or became verbally insulting, swore, or shouted. Also, 80% became physically aggressive, such as slamming doors, punching walls, making a mess, or destroying property, and 58% hit, kicked, bit, or spit, and 19% needed physical restraint. Moreover, 79% had outbursts at least weekly, and 58% of those lasted for half an hour or more. In comparison, at the end of the trial, 80% still argued, whined or sulked, and but only 31% became verbally insulting, swore, or shouted. Further, only 31% became physically aggressive, 29% hit, kicked, bit or spit, and 10% required physical restraint. A Table displaying detailed EMO-I outcomes is shown in the supplemental material (S2 Table).

## Adverse events, adherence rates, and product likability

Across the whole study period, 168 side effects were reported while taking the micronutrient Sticks, and the caregiver believed that 29% ($N = 48$) of those were linked to the intervention. No serious adverse events were reported. Fifty-one percent ($N = 85$) either persisted for at least four weeks or reoccurred throughout the trial. At the end of RCT, the IWLG reported experiencing significantly more headaches ($\chi^2(3) = 8.29$, $p = .040$) and sweating ($\chi^2(1) = 4.36$, $p = .037$) than the ITG; in particular, headaches were experienced by 50% ($N = 12$) of the

IWLG and 25% ($N = 6$) of the ITG, and sweating by 17% ($N = 4$) and 0% ($N = 0$) in the IWLG and ITG, respectively. Throughout the trial, increased appetite was the most frequently reported side effect linked by parents to the intervention, although nausea and vomiting was the most reported adverse event overall. For more information on overall side effects, see Table 4. Side effects during the RCT phase can be found in the supplemental material (S3 Table).

Based on self-report, the adherence rate was 92% in the ITG and 94.9% in the IWLG (over-all 93%). The caregivers rated it to be 'slightly easy' ($M = 5.4$, $SD = 1.4$) for the children to take the product and reported that children liked the micronutrient powder 'a little' ($M = 4.8$, $SD = 1.4$). 44% of the participants liked banana, 32% tropical punch, and 28% sour berry the most. Most caregivers of participants (95%) reported they would recommend the micronutrient product and 87% would purchase it at some point in the future. The main reason reported for not wanting to buy it in the future was cost.

## Discussion

This study presents the first reported trial to investigate the effects of micronutrients absorbed by the oral mucosa on emotion dysregulation symptoms in 5-10-year-old children. By the end of the RCT phase, emotion dysregulation and overall improvement were significantly greater in the ITG than in the IWLG. In addition, the ITG improved on the ODD subscale of the

**Table 4. Treatment-emerging adverse events.**

| Adverse events | Participants ($N = 48$) (%) | Linked to intervention[a] ($N = 48$) (%) | Persistent[b] and/or recurrent ($N = 48$) (%) |
|---|---|---|---|
| Dry mouth | 7 (14.6) | 2 (4.2) | 4 (8.3) |
| Drowsiness | 9 (18.8) | 1 (2.1) | 5 (10.4) |
| Insomnia | 11 (22.9) | 7 (14.6) | 10 (20.8) |
| Blurred vision | 3 (6.3) | 0 (0) | 0 (0) |
| Headache | 13 (27.1) | 3 (6.3) | 9 (18.8) |
| Constipation | 3 (6.3) | 2 (4.2) | 0 (0) |
| Diarrhea | 14 (29.2) | 1 (2.1) | 3 (6.3) |
| Increased appetite | 17 (35.4) | 12 (25) | 14 (29.2) |
| Decreased appetite | 11 (22.9) | 4 (8.3) | 5 (10.4) |
| Nausea and vomiting | 20 (41.7) | 4 (8.3) | 9 (18.8) |
| Problems with urination | 3 (6.3) | 0 (0) | 2 (4.2) |
| Palpitations | 2 (4.2) | 1 (2.1) | 1 (2.1) |
| Feeling light-headed on standing | 6 (12.5) | 2 (4.2) | 2 (4.2) |
| Feeling like the room is spinning total | 11 (22.9) | 1 (2.1) | 1 (2.1) |
| Sweating | 7 (14.6) | 1 (2.1) | 5 (10.4) |
| Increased body temperature | 8 (16.7) | 2 (4.2) | 3 (6.3) |
| Tremor | 0 (0) | 0 (0) | 0 (0) |
| Disorientation | 1 (2.1) | 0 (0) | 0 (0) |
| Yawning | 11 (22.9) | 1 (2.1) | 3 (6.3) |
| Weight gain | 8 (16.7) | 4 (8.3) | 6 (12.5) |
| Eczema | 1 (2.1) | 2 (4.2) | 3 (6.3) |

*Note.*

[a]Based on parent-report and clinical judgement.

[b]Persistent = present for at least four weeks.

SNAP-IV to a significantly larger degree than the IWLG and there was a trend towards a greater improvement on the ADHD subscale. The reduction in symptoms was clinically meaningful in the ITG, reducing ADHD scores from the mild to the non-clinical range, whereas the IWLG stayed within the mild range. Similarly, on the ODD subscale, scores in the ITG reduced from the moderate to the mild range, while the IWLG stayed within the moderate range.

Comparing the results from the current study to CBT-based interventions on emotion regulation, micronutrients reduced symptoms to a larger degree. In particular, a meta-analysis by Mingebach, Kamp-Becker [58] found that parent-based psychological interventions for children under the age of 13 yielded overall long-term moderate improvements in children's externalizing (*SMD* = 0.45, *p* < .001) and overall behavior (*SMD* = 0.46, *p* < .001). Further, Kennedy, Bilek [59] found that a transdiagnostic intervention targeting areas related to emotional reactivity and anxiety-focused CBT in children with emotional disorders led to approximately 70% of treatment response over 15 and 10 sessions respectively. These findings are comparable to the current study, where 65% of children in the ITG had responded to treatment; however, micronutrients had only been administered for four weeks at this point, showing a more rapid effect. In a recent randomized controlled study by Helander, Enebrink [60], 120 8-12-year old children with ODD, ODD combined with CD or Disruptive Behavior Disorder Not Otherwise Specified (NOS) were assigned to 11 group sessions of parent treatment alone or combined with group-CBT for children. Both treatment groups showed a large decrease in symptoms related to ODD from pre-treatment to 2-year follow-up (*d* = -0.80, *p* < .001). Comparing these findings to our results, micronutrients have a more immediate and potentially larger effect on emotion dysregulation symptoms in children. In addition, micronutrients are convenient to administer, require less time, effort, and money both by parents and children, and place fewer demands on health professionals and the broader health system.

Regarding psychiatric medication, antipsychotics have been shown to have a positive impact on anger and disruptive behavior related to psychiatric illnesses in children and adults [61, 62]; nevertheless, concerns have been raised regarding significant adverse side effects, and in most cases, medication alone does not suffice for children with severe emotion dysregulation and must be administered in addition to CBT-based interventions (n.b. the same may potentially be true for micronutrients) [62, 63]. A recent prospective naturalistic study investigated the effect of low and high dose PRN (as-needed) psychiatric medication on anger outbursts in 104 children in a psychiatric hospital, where children did not show any improvements in outburst duration [64]. Our findings, in contrast, showed that at the end of the trial, outbursts became less severe, less frequent, and shorter in duration compared to baseline.

Our positive findings are underlined by previous evidence illustrating that micronutrients can alleviate emotion dysregulation in children and adults. For example, a double-blind RCT by Rucklidge, Eggleston [29] investigated the effects of taking a broad-spectrum micronutrient supplement for 10 weeks for children with ADHD compared to a placebo. Results revealed that relative to the control group, the micronutrients moderately improved symptoms of emotion dysregulation (*d* = 0.46, *p* = .029). In addition, 47% of the micronutrient group overall 'much' or 'very much' improved, while only 28% did so in the placebo group. Treatment response rates were slightly higher in our study, where 67% in the ITG 'much' or 'very much improved' at the end of RCT, and 59% at the end of the trial. Nevertheless, it is worth noting that the Rucklidge, Eggleston [29] study was blinded compared to our non-blinded study, representing a possible explanation for the lowered improvement rates, as non-blinded studies can overestimate treatment effects.

Further supporting our findings, a recent study with 126 children with ADHD aged 6–12 found that at week 8, 38% 'definitely-', 'much-', or 'extremely improved' in the micronutrient compared to 25% in the placebo group ($p$ = .04) [65]; however, regarding emotion dysregulation and irritability specifically, no group difference was found. This stands in contrast to the current study, where regarding emotion dysregulation-related symptoms, 67% 'definitely improved' or better in the ITG compared to 8% in the IWLG.

Additionally, a positive correlation between scores on the Total CL-ARI and Total DASS-21 at the end of RCT phase was found, indicating that the lower the participants' symptoms of irritability, the lower the caregivers' symptoms of depression, anxiety, and stress, and vice versa. Further, the beneficial effects of the initial exposure to micronutrients did not fully wash out during the first OFF phase for either group. This may be related to the direct effects of the micronutrients, or it might be related to more positive interactions between child and parent, where potential changes in the relationship that occurred when behavior improved in the ON phase were maintained to an extent when coming off the micronutrients. This highlights an important connection between the caregiver's and children's mental health, which has also been shown in previous research; for example, Karimzadeh, Rostami [66] found that different aspects of the parent's functioning (physical and social functioning, anxiety, and depression) were positively correlated with externalizing and internalizing symptoms in their preschool children, including aggression, ignorance, and symptoms of anxiety.

During the maintenance phase, individuals in the ITG showed a significantly greater overall improvement on the CGI-I than the IWLG, and in addition, more individuals from the IWLG withdrew from the study compared to the ITG (15% ($N$ = 5) and 10% ($N$ = 2) respectively). A previous study found that length of time spent on a waitlist predicted treatment dropout among individuals with an eating disorder [67]. As motivation to engage in treatment is likely at its maximum when first seeking help, and may weaken over time, waiting might result in diminished commitment to treatment [67]. Although it was observed that no participants dropped out after the 4-week waiting period, their motivation to stay in the study might have been weakened throughout the waiting period, leading to more dropouts later in the study. Additionally, waitlist participants were in the trial for four weeks longer than the treatment group, which may also have contributed. Overall, this suggests some disadvantages for participants who are placed on a waitlist.

With regards to treatment-emergent side effects, at the end of RCT, the IWLG reported significantly more headaches and sweating than the ITG. Overall, an increase in appetite was the most frequently reported treatment-emergent side effect considered to be related to the intervention. These findings are consistent with previous research finding increased food consumption following nutritional supplementation [68–70].

The overall dropout rate was low (14.5%) and adherence to the intervention was high. Further, participants reported that they liked the micronutrients 'a little' and that they were 'slightly easy' to take. Our results confirm previous findings by Katta, Blampied [33], indicating that the micronutrient Sticks are a convenient mode of taking micronutrients.

Overall, our results show micronutrient Sticks lead to significant improvements in emotional dysregulation in children, improvements in ODD symptoms and parental functioning, and had an appealing side effect profile. Although future trials and replication of these findings is necessary to strengthen and better understand the impact of micronutrients absorbed via the oral mucosa on emotion dysregulation, the results from this study indicate that they might be an effective treatment for children with moderate-to severe symptoms of emotion dysregulation.

## Limitations and future directions

All online measures were completed by the participants' caregivers, and no one was blinded to the intervention or waitlist group allocation. This introduces potential bias due to expectancy effects, including performance bias, as knowing about taking an active treatment is likely to influence participants' expectations. Moreover, we recognize that compared to active control groups, waitlist participants improve less or not at all, as receiving no treatment may interfere with potential spontaneous symptom alleviation by reinforcing participant's belief that they will not improve [71]. This can contribute to an overestimated treatment effect.

Furthermore, it needs to be underlined that the current findings are exclusively based on samples recruited from New Zealand. Larger future studies including samples from a variety of geographical backgrounds need to be conducted to support the efficacy of the intervention among other populations.

In addition, the study findings established some disadvantages of being placed in the wait-list group in terms of treatment effectiveness and completion, which are ethical issues that need to be considered in future research. As our trial was exploratory in nature, future double-blinded randomized trials need to address these issues to strengthen internal validity.

As past research has established a dose-response curve for the micronutrients used in the Sticks [72], it would be interesting to look into the effects of an increased Stick dose on emotion dysregulation in future studies, especially in those who did not show a clinical response to the initial dose. It might also be useful to further investigate the potential relationship between micronutrient supplement intake and appetite in children that was observed in this trial. Lastly, future studies should explore the pharmacokinetics of oral mucosal and gastrointestinal micronutrient administration to evaluate which absorption method might be more effective [33].

## Supporting information

**S1 Table. Ingredients and doses of the micronutrients.**
(DOCX)

**S2 Table. Descriptive information for the EMO-I for participants at baseline and end of trial.**
(DOCX)

**S3 Table. Recorded adverse events during the RCT phase.**
(DOCX)

**S1 Fig. Study phases.**
(TIF)

**S1 Appendix. CONSORT 2010 checklist.**
(DOC)

**S2 Appendix. Study protocol for micronutrients and emotion dysregulation in children.**
(DOCX)

## Acknowledgments

Thanks go to the participants and their families, and to Truehope for donating the micronutrient product researched in this study (Truehope Ultimate Sticks).

## Author Contributions

**Conceptualization:** Nurina M. Katta, Neville M. Blampied, Julia J. Rucklidge.

**Data curation:** Nurina M. Katta.

**Formal analysis:** Nurina M. Katta, Neville M. Blampied.

**Investigation:** Nurina M. Katta, Julia J. Rucklidge.

**Methodology:** Nurina M. Katta, Neville M. Blampied, Julia J. Rucklidge.

**Project administration:** Nurina M. Katta, Neville M. Blampied.

**Resources:** Nurina M. Katta.

**Software:** Nurina M. Katta, Neville M. Blampied.

**Supervision:** Neville M. Blampied, Matt Eggleston, Julia J. Rucklidge.

**Visualization:** Nurina M. Katta.

**Writing – original draft:** Nurina M. Katta.

**Writing – review & editing:** Nurina M. Katta, Neville M. Blampied, Julia J. Rucklidge.

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
