## [Decision Letter · Decision Letter 0]

19 Jun 2024

PONE-D-24-09885Micronutrients absorbed via the oral mucosa reduce emotion dysregulation in 5-10-year-old children: A three-phased randomized wait-list-controlled trialPLOS ONE

Dear Dr. Rucklidge,

Thank you for submitting your manuscript to PLOS ONE. After careful consideration, we feel that it has merit but does not fully meet PLOS ONE’s publication criteria as it currently stands. Therefore, we invite you to submit a revised version of the manuscript that addresses the points raised during the review process.

We look forward to receiving your revised manuscript.

Kind regards,

Ebenezer Wiafe, PhD, MPharm, Pharm D

Academic Editor

PLOS ONE

Journal Requirements:

Reviewers' comments:

Reviewer's Responses to Questions

**Comments to the Author**

1. Is the manuscript technically sound, and do the data support the conclusions?

Reviewer #1: Partly

Reviewer #2: Yes

Reviewer #3: Yes

2. Has the statistical analysis been performed appropriately and rigorously? 

Reviewer #1: No

Reviewer #2: Yes

Reviewer #3: Yes

3. Have the authors made all data underlying the findings in their manuscript fully available?

Reviewer #1: No

Reviewer #2: Yes

Reviewer #3: Yes

4. Is the manuscript presented in an intelligible fashion and written in standard English?

Reviewer #1: Yes

Reviewer #2: Yes

Reviewer #3: Yes

5. Review Comments to the Author

Reviewer #1: This manuscript presents data analysis from a 3-phase, randomized, open-label, waitlist-controlled trial to compare

the effectiveness of orally-absorbed micronutrients intervention in 5-10 year old children. The topic is of importance, and the study was registered as a RCT, and was approved by the respective IRB/Ethics Committee. While the study objectives sound interesting, is important, and on target, some shortcomings were observed, in regards to abiding by the CONSORT guidelines for conducting and reporting results of high-quality randomized controlled trials (RCTs). Some other (statistical) comments were also provided.

1. Abstract:

For a better presentation, Abstract needs to follow the Objective/Methods/Results/Conclusion framework.

2. Methods:

Methods reporting need some work. An orderly manner is suggested, following CONSORT guidelines, without repeating information, such as Trial Design, Participant Eligibility Criteria and settings, Interventions, Outcomes, sample size/power considerations, Interim analysis and stopping rules, Randomization (details on random number generation, allocation concealment, implementation), Blinding issues, etc, should be mentioned. The authors are advised to create separate subsections for each of the possible topics (whichever necessary), and that way produce a very clear writeup. They are advised to write it carefully, following nice examples in the manuscript below:

https://www.sciencedirect.com/science/article/pii/S0889540619300010

Specific comments:

(a) For instance, the randomization and allocation concealment should be made very clear (they are NOT the same thing); the trial staff recruiting patients should NOT have the randomization list. Randomization should be prepared by the trial statistician, and he/she would not participate in the recruiting.

(b) More details on the randomization is needed; saying "a computer-generated randomization" was used is half-hearted.

(c) Sample size: The sample size/power statement appeared pretty vague; in usual waitlist-controlled trials, there are previous attempts of producing a desirable sample size/power statements, using the ANCOVA design; see here:

https://www.ncbi.nlm.nih.gov/pmc/articles/PMC5109669/ The authors are recommended to do the same. If not possible, then some power/sample size statements should be produced, based on an objective effect size, and at a reasonable 5% level of significance.

(d) Statistical Analysis:

(d1) The section Data Analysis should be renamed Statistical Methods. While mentioning tests, whether a 1-sided, or a 2-sided test will be used needs to be mentioned. Furthermore, t-test was used (for continuous variables) which is heavily based on normality assumptions. Under violations, alternative nonparametric tests, such as Wilcoxon rank sum tests, should be mentioned. There is no mention of such alternative testing methods.

(d2) ANOVA was used, however, it is once again heavily based on Gaussian assumptions. No nonparametric alternatived to ANOVA mentioned.

(d3) Is there any reason why a ANCOVA was not used, which is typically the model for waitlist-controlled trials, incorporating covariates.

3. Results & Conclusions:

(a) The authors should check that any statement of significance should be followed by a p-value in the entire Results section. Otherwise, the Results section look OK; it's pretty straightforward.

(b) Conclusions should state that the current findings are ONLY based on the random samples derived from a New Zealand population, and should allude to future studies with much larger sample sizes and collected at other geographical areas to confirm the effectiveness of the intervention.

Reviewer #2: This is a well-written manuscript that contributes to the literature on the use of alternative therapy in emotional dysregulation/ADHD symptom amelioration.

Below are some suggestions for revisions/clarifications:

1. The study’s justification for focusing on children ages 5-10 yr old, need to be made clear. One possible assumption is that it is because this age group has difficulty with pill swallowing. This needs to be clarified in the write-up.

2. It is suggested that the authors clarify the level of involvement of supplement providers with regards to study design, implementation, reporting, etc.

3. CL-ARI is supposed to be administered by a clinician. It is not clear if the study coordinator is a clinician and was therefore qualified to administer this measure. Also, it is not clear if the PTP and CGI-S were completed by clinicians or the research coordinator.

4. Similar to the above comment, it is not clear who completed the CGI and how the determination of treatment response was made.

5. There are about 10 secondary outcomes. What was the justification for assessing all of these? For example, it is not clear why the EBQ was assessed since the study was not targeted at improving diet.

6. Were the reported PTPs similar for all participants? If not, it is not clear how the individual PTPs were classified and grouped to determine symptom improvements per group (i.e ITG vs. IWLG)?

7. The assessment of safety is one of the study goals , however, the methods is missing how safety was operationalized and assessed.

8. Figure ( esp 2 & 3) titles/notes are separated from the actual figures (pages 15 vs. 47/48), which made comprehending the tables difficult.

9. Suggest minimizing reporting results in discussion.

Reviewer #3: ### Review Comments for Manuscript PONE-D-24-09885

#### 1. Technical Soundness and Data Support

The manuscript is technically sound and presents a well-conducted study on the effects of micronutrients absorbed via the oral mucosa on emotion dysregulation in children. The study design, including the randomized wait-list-controlled trial and three-phased approach, is robust and appropriate for the research question. The data support the conclusions, demonstrating significant improvements in emotion dysregulation for children receiving the intervention compared to the control group. The study also includes appropriate controls, replication, and sample sizes to ensure the validity of the findings.

#### 2. Statistical Analysis

The statistical analysis is performed rigorously and appropriately. The authors use a range of statistical tests, including t-tests, ANOVA, and effect size calculations, to analyze the data. The reporting of results includes confidence intervals and effect sizes, providing a clear understanding of the magnitude and significance of the findings. The use of Modified Brinley Plots to display individual changes over time is a valuable addition that enhances the interpretation of the data.

#### 3. Data Availability

The authors have made a commendable effort to make the data underlying the findings available, in line with PLOS ONE's data policy. While the raw data are not included in the manuscript to protect participant privacy, the authors have provided a clear statement that de-identified data will be available upon reasonable request. This approach balances the need for data transparency with ethical considerations regarding participant confidentiality.

#### 4. Manuscript Presentation and Language

The manuscript is presented in an intelligible fashion and is written in clear, standard English. The structure is logical, and the flow of information is easy to follow. There are no significant typographical or grammatical errors, and the language is appropriate for a scientific audience.

#### 5. Review Comments to the Author

- The manuscript is a valuable contribution to the field of child psychology and nutrition, addressing a relevant and timely research question.

- The introduction provides a thorough background, highlighting the need for alternative treatments for emotion dysregulation in children and the potential of micronutrient interventions.

- The methodology is well-detailed, ensuring that the study can be replicated. The randomization process, intervention details, and measurement tools are clearly described.

- The results are presented comprehensively, with appropriate use of tables and figures to illustrate key findings. The inclusion of both primary and secondary outcome measures strengthens the study.

- The discussion section effectively interprets the findings, placing them in the context of existing literature. The authors acknowledge the limitations of the study, including the open-label design and potential biases, and suggest directions for future research.

- The ethical considerations and adherence to data availability policies are well-handled, demonstrating a commitment to ethical research practices.

6. PLOS authors have the option to publish the peer review history of their article (what does this mean?). If published, this will include your full peer review and any attached files.

Reviewer #1: No

Reviewer #2: No

Reviewer #3: **Yes: **sherif kamal

---

## [Author Response · Author response to Decision Letter 0]

11 Aug 2024

PLOS One response letter

Reviewer #1:

1. Abstract

Comment: For a better presentation, the Abstract needs to follow the Objective/Methods/Results/Conclusion framework.

Response: The Abstract has been adjusted to the Objective/Methods/Results/Conclusion framework.

2. Methods: 

Comment: Methods reporting need some work. An orderly manner is suggested, following CONSORT guidelines, without repeating information, such as Trial Design, Participant Eligibility Criteria and settings, Interventions, Outcomes, sample size/power considerations, Interim analysis and stopping rules, Randomization (details on random number generation, allocation concealment, implementation), Blinding issues, etc, should be mentioned. The authors are advised to create separate subsections for each of the possible topics (whichever necessary), and that way produce a very clear writeup. They are advised to write it carefully, following nice examples in the manuscript below:

https://www.sciencedirect.com/science/article/pii/S0889540619300010

A) For instance, the randomization and allocation concealment should be made very clear (they are NOT the same thing); the trial staff recruiting patients should NOT have the randomization list. Randomization should be prepared by the trial statistician, and he/she would not participate in the recruiting.

B) Comment: More details on the randomization is needed; saying "a computer-generated randomization" was used is half-hearted.

Response to comments A and B: The Methods section has been adjusted to the suggested format and the randomization process has been clarified on p. 13: “Participants were randomized in blocks of four to the initial treatment and initial waitlist control group (ITG and IWLG respectively) using a computer-generated randomization sequence (from www.randomization.com) by an independent person not involved in the recruitment of participants. The website asked about the set of numbers that needed to be generated, the numbers per set, and the number range. Once the randomization list had been generated, participants were assigned to the next sequential number that had been written on a paper and concealed in envelopes by an independent person not involved in the recruitment process. During the baseline meeting, the envelopes were opened by the study coordinator in front of the participants, who had been blinded to participant randomization and assignment up to this point.” 

C) Comment: Sample size: The sample size/power statement appeared pretty vague; in usual waitlist-controlled trials, there are previous attempts of producing a desirable sample size/power statements, using the ANCOVA design; see here:

https://www.ncbi.nlm.nih.gov/pmc/articles/PMC5109669/ The authors are recommended to do the same. If not possible, then some power/sample size statements should be produced, based on an objective effect size, and at a reasonable 5% level of significance.

Response: The sample size/power statement was adjusted and made more specific (p. 5). “The number of participants was based on the What Works Clearinghouse Standards for single-case designs (37, 38), being considered a sufficient sample size to determine whether the treatment is successful in reducing emotion dysregulation with sufficient power at a 5% level of significance. The researchers anticipated a moderate effect size (~d=0.5) based on previous studies investigating the same intervention (i.e. True Hope Ultimate Sticks), establishing moderate effect sizes on emotion dysregulation (33).”

The design standard for single-case experimental designs is based on validity considerations following the What Works Clearinghouse Standards. As the study employed an initial RCT phase followed by a replication crossover withdrawal design drawn from a single-case ABAB design, the number of participants was based on the What Works Clearinghouse Standards for single-case designs, being considered a sufficient sample size to determine whether the treatment is successful in reducing emotion dysregulation with sufficient power. Therefore, no conventional power calculation was conducted. The anticipated effect size was based on findings on emotion dysregulation from previous research (Katta et al., 2023) investigating the same product as in the current study.

D) Statistical analysis

Results and conclusions

(d1) Comment: The section Data Analysis should be renamed Statistical Methods. While mentioning tests, whether a 1-sided, or a 2-sided test will be used needs to be mentioned. Furthermore, t-test was used (for continuous variables) which is heavily based on normality assumptions. Under violations, alternative nonparametric tests, such as Wilcoxon rank sum tests, should be mentioned. There is no mention of such alternative testing methods.

Response: A two-sided t-test was used. Because our design is balanced, with equal n in each group, and additionally is an RCT, our design is particularly robust against violations of parametric test assumptions (Blanca, 2018; Schmider et al., 2010). Nevertheless, we tested for normality using the Shapiro-Wilk Test, and for all variables that failed the normality test at baseline, the data was analysed using the Welch’s Test. The Welch’s Test is a parametric test that controls for homogeneity of variance, and in addition, it delivers an effect size. The results were consistent with the original findings. All this information was added to the Data Analysis section (p. 12) and adjusted in the Results section (p. 13-23).

(d2) ANOVA was used, however, it is once again heavily based on Gaussian assumptions. No nonparametric alternatives to ANOVA mentioned.

Response: ANOVA is widely accepted to be robust from normality when sample sizes are relatively large (Mena et al., 2017, Schmider et al., 2010). In addition, our design is robust against test assumption violations. Therefore, we decided that data analysis with a nonparametric alternative was not necessary. 

(d3) Is there any reason why an ANCOVA was not used, which is typically the model for waitlist-controlled trials, incorporating covariates. 

Response: The study employed an initial RCT phase followed by a replication crossover design. At the end of each replication phase, there we compared two groups at one time point; therefore, the t-test appeared to be a valid and sufficient way of analyzing the data. This is supported by Wilkinson (1999), the Task Force on Statistical Inference, stating that a minimally sufficient analysis should be chosen to provide a simple yet adequate way to address the data. Further, the variance of scores was initially compressed using strict eligibility criteria, reducing the occurrence of a large variance in scores. 

Nevertheless, we re-analyzed the RCT data using ANCOVA with the baseline data as a covariate, and the results did not change. Therefore, we decided to keep reporting the outcomes using t-tests. 

Results & Conclusions:

A) Comment: The authors should check that any statement of significance should be followed by a p-value in the entire Results section. Otherwise, the Results section look OK; it's pretty straightforward.

Response: The authors checked each statement of significance and ensured that each statement was followed by a p-value.

B) Conclusions should state that the current findings are ONLY based on the random samples derived from a New Zealand population, and should allude to future studies with much larger sample sizes and collected at other geographical areas to confirm the effectiveness of the intervention.

Response: This has been added to the Limitations and Future Directions section (p. 30): “Furthermore, it needs to be underlined that the current findings are exclusively based on samples recruited from New Zealand. Larger future studies including samples from a variety of geographical backgrounds need to be conducted to support the efficacy of the intervention among other populations.” 

Reviewer #2:

1. Comment: The study’s justification for focusing on children ages 5-10 yr old, need to be made clear. One possible assumption is that it is because this age group has difficulty with pill swallowing. This needs to be clarified in the write-up.

Response: The justification for focusing on 5- to 10-year-old children has been clarified on pages 3-5 in the introduction. One reason for focusing on this age group is that emotion regulation abilities typically develop during the first 5 years of life and represent normative behavior in early childhood/preschool age, which needs to be differentiated from clinically significant symptoms. Another reason is that we wanted to minimize confounding variables as best as possible, for example developmental levels (e.g. puberty). Further, some of the measures used, such as the SDQ, are applicable to an age range from 4-10 years. Finally, children often struggle with swallowing pills. 

2. Comment: It is suggested that the authors clarify the level of involvement of supplement providers regarding the study design, implementation, reporting, etc.

Response: The supplement providers were not involved in any stage of the study design, including study planning, design, implementation, or reporting. This has been added on p. 8.

3. CL-ARI is supposed to be administered by a clinician. It is not clear if the study coordinator is a clinician and was therefore qualified to administer this measure. Also, it is not clear if the PTP and CGI-S were completed by clinicians or the research coordinator.

Response: The CL-ARI was administered by the study coordinator, a postgraduate PhD student in Psychology and supervised by a Clinical Psychologist. As part of the supervision, the supervisor conducted reliability checks and assessed whether the CL-ARI procedure was followed correctly. This has been added to the methods section.

4. Comment: Similar to the above comment, it is not clear who completed the CGI and how the determination of treatment response was made.

Response: The CGI was completed by the study coordinator, a postgraduate PhD student in Psychology who was supervised by a Clinical Psychologist. Treatment response was determined based on all information that was available to the study coordinator, including all parent-report and clinician-administered measures. This has been added to the methods section.

5. There are about 10 secondary outcomes. What was the justification for assessing all of these? For example, it is not clear why the EBQ was assessed since the study was not targeted at improving diet.

Response: A range of secondary outcomes were assessed to ensure a comprehensive picture of symptoms related to emotion dysregulation. As emotion dysregulation can present in various ways, and there are inconsistencies in its definition in the literature, it was important to include multiple emotion dysregulation measures. 

The EBQ was assessed to be able to measure whether the participant’s diet changed over time and being able to exclude that diet might have confounded the results. 

The DASS-21 was related to the caregiver’s wellbeing to assess a potential connection between the children’s symptoms of emotion dysregulation and the caregiver’s wellbeing, or vice versa.

6. Were the reported PTPs similar for all participants? If not, it is not clear how the individual PTPs were classified and grouped to determine symptom improvements per group (i.e ITG vs. IWLG)? 

Response: The analyzed PTPs were related to emotion dysregulation, including temper outbursts, emotion dysregulation, frustration (outbursts), anger (outbursts), and emotional outbursts. Given the limited sample size, other PTPs not related to emotion dysregulation were not categorized as these largely varied between participants. 

7. The assessment of safety is one of the study goals , however, the methods is missing how safety was operationalized and assessed.

Response: The safety was assessed by capturing side effects with an established side effect measure (the Revised Side-Effect Checklist). This has been added to the Measures section under Methods (p. 12). 

8. Figure (esp 2 & 3) titles/notes are separated from the actual figures (pages 15 vs. 47/48), which made comprehending the tables difficult.

Response: We followed the PLOS One formatting guidelines which outlined to only insert the figure titles in the manuscript. 

9. Suggest minimizing reporting results in discussion.

Response: Statistical results that had already been mentioned previously in the results were removed from the discussion.

Reviewer #3:

Response: Thank you for the very positive reflection, it is much appreciated.

---

## [Decision Letter · Decision Letter 1]

25 Sep 2024

Micronutrients absorbed via the oral mucosa reduce emotion dysregulation in 5-10-year-old children: A three-phased randomized wait-list-controlled trial

PONE-D-24-09885R1

Dear Dr. Rucklidge,

We’re pleased to inform you that your manuscript has been judged scientifically suitable for publication and will be formally accepted for publication once it meets all outstanding technical requirements.

Kind regards,

Ebenezer Wiafe, PhD, MPharm, Pharm D

Academic Editor

PLOS ONE

---

## [Editor Report · Acceptance letter]

18 Oct 2024

PONE-D-24-09885R1 

PLOS ONE

Dear Dr. Rucklidge, 

I'm pleased to inform you that your manuscript has been deemed suitable for publication in PLOS ONE. Congratulations! Your manuscript is now being handed over to our production team.

Kind regards, 

on behalf of

Dr. Ebenezer Wiafe 

Academic Editor

PLOS ONE